# Piperlongumine Analogs Promote A549 Cell Apoptosis through Enhancing ROS Generation

**DOI:** 10.3390/molecules26113243

**Published:** 2021-05-28

**Authors:** Ai-Ling Sun, Wen-Wen Mu, Yan-Mo Li, Ya-Lei Sun, Peng-Xiao Li, Ren-Min Liu, Jie Yang, Guo-Yun Liu

**Affiliations:** 1School of Chemistry and Chemical Engineering, Liaocheng University, Liaocheng 252059, China; sunailing1985@126.com; 2School of Pharmaceutical Sciences, Liaocheng University, Liaocheng 252059, China; MWW1115@163.com (W.-W.M.); Lipx19986@163.com (P.-X.L.); liurenmin@lcu.edu.cn (R.-M.L.); 3Shandong Center for Disease Control and Prevention, Jinan 250014, China; liyanmo163@163.com; 4Qingdao Vland Biotech INC, Qingdao 266000, China; sunyl01@vlandgroup.com

**Keywords:** piperlongumine an alog, michael acceptor, anticancer activity, reactive oxygen species, thioredoxin reductase

## Abstract

Chemotherapeutic agents, which contain the Michael acceptor, are potent anticancer molecules by promoting intracellular reactive oxygen species (ROS) generation. In this study, we synthesized a panel of **PL** (piperlongumine) analogs with chlorine attaching at C2 and an electron-withdrawing/electron-donating group attaching to the aromatic ring. The results displayed that the strong electrophilicity group at the C2–C3 double bond of PL analogs plays an important role in the cytotoxicity whereas the electric effect of substituents, which attached to the aromatic ring, partly contributed to the anticancer activity. Moreover, the protein containing sulfydryl or seleno, such as TrxR, could be irreversibly inhibited by the C2–C3 double bond of **PL** analogs, and boost intracellular ROS generation. Then, the ROS accumulation could disrupt the redox balance, induce lipid peroxidation, lead to the loss of MMP (Mitochondrial Membrane Potential), and ultimately result in cell cycle arrest and A549 cell line death. In conclusion, **PL** analogs could induce in vitro cancer apoptosis through the inhibition of TrxR and ROS accumulation.

## 1. Introduction

Natural products have been markedly studied for drug discovery, including anti-inflammatory agents [1], anti-bacterial agents [2], and particularly anti-cancer agents [3,4]. Piperlongumine (**PL**), a natural alkaloid isolated from traditional Chinese medicine *Piper longum* L., is well known to possess multiple pharmacological effects, such as anti-microbial [5], anti-Parkinsonian [6] and anti-epileptic [7] effects. So far, hundreds of publications have shown that **PL** could exhibit anticancer activities in various cancer cells [8,9], and effectively inhibit tumor growth without considerable side effects in tumor xenograft models [10]. Moreover, **PL** induces cell cycle arrest and death of A549 cells through triggering reactive oxygen species (ROS) accumulation in cancer cells [11,12].

ROS, well known as an upstream signal involved in the induction of cancer cell death, can be triggered by several exogenous and endogenous factors [13,14]. ROS induced by chemotherapeutic agents may be mainly due to the inhibition of the antioxidant system. For example, curcumin [15] and **PL** [16], characterized with the Michael acceptor, could irreversibly inhibit thioredoxin reductase (TrxR), and the adduct triggers ROS generation. The electrophilicity of Michael acceptor units is important for the TrxR inhibition and ROS generation ability [17]. Yan and co-workers have designed piperlongumine-directed anticancer agents by an electrophilicity-based prooxidant strategy [16]. The structure–activity relationship of **PL** analogs reveals that C2–C3 and C7–C8 double bonds are essential for their cytotoxicity against cancer cells [18] (Scheme 1). C2–C3 double bond plays a critical role in the ROS generation and glutathione lessening whereas C7–C8 double bonds are partially correlated with the anti-proliferative activity of **PL**, and the effect of the pharmacophore on anticancer activity remains unexplored.

Additionally, fluorine, which usually presents in drugs, exihits a variety of properties to certain medicines, such as increased metabolic stability, binding interactions, selective reactivities and so on [19]. Considering the aforementioned, we synthesized a panel of PL analogs with electron-withdrawing group (-F and CF_3_) or electron-donating group (-OCH_3_) attaching to the aromatic ring (Scheme 1). Then, we studied the electronic effect of substituents on the cytotoxicity and explored the anticancer mechanism of **PL** analogs against A549 cells.

## 2. Results

### 2.1. Chemistry

The synthetic routes of **PL** analogs are illustrated in Scheme 1. The intermediate **4** (3-chloro-5,6-dihydropyridin-2(1*H*)-one) was obtained from commercial 2-piperidone through two steps, including α-halogenation and elimination. Then, the appropriate cinnamoyl chloride **2**, condensing between benzaldehyde and malonic acid, reacted with intermediate **4** through a nucleophilic substitution reaction to afford the target comounds with lower yields. All compounds were characterized by nuclear magnetic resonance (^1^H-NMR) and ^13^C-NMR.

### 2.2. Assessment of In Vitro Cytotoxicity

**PL** analogs **1**–**6** were determined for their cytotoxicity aganinst A549 and SKOV3 cells by MTT (3-(4,5-Dimethylthiazol-2-yl)-2,5-diphenyltetrazolium bromide) assay. As illustrated in Table 1, most of the **PL** analogs were more excellent or equivalent to **PL** against cancer cells, and the cytotoxicity order as the follows: **PL-6** > **PL-1** > **PL** ~ **PL-5** ~ **PL-2**. This result indicated that the electron-withdrawing group (chlorine) attached to C2 could strengthen the antiproliferative activity of PL analogs. Moreover, the outstanding cytotoxicity of **PL-2**, **PL-5** and **PL-6** may be attributed to the electron-withdrawing effect of -F and -CF_3_, which could heighten the electrophilicity of the C7–C8 double bond. Additionally, **PL-2**, **PL** analogs were stable in phosphate-buffered saline (PBS) buffer at 25 °C (Figure 1). **PL-1** and **PL-6** showed the strongest antiproliferative activity against A549 cells among these compounds and, therefore, we focused on A549 cells to investigate the cytotoxic mechanism of **PL-1** and **PL-6** in subsequent experiments.

### 2.3. Piperlongumine (PL) Analogs Triggered Cell Cycle Arrest

In order to explore the effects of **PL** analogs (**PL-1** and **PL-6**) on cell cycle arrest, flow cytometry was used to determine the cell cycle distribution. As shown in Figure 2, **PL-1** and **PL-6** obviously reduced the percentage of cells in G0/G1 phase, and enhanced the percentage of cells in G2/M phase in a dose-dependent fashion, relative to the control group. Moreover, **PL-1** and **PL-6** were excellent or equivalent to **PL** on the cell cycle arrest of A549 cells; 30 μM **PL-6** led 36.4% percentage of cells in the G2/M phase. In addition, the classic antioxidant-NAC (*N*-Acetyl-*L*-cysteine) could effectively inhibit the cell cycle arrest activity induced by **PL-1** and **PL-6**. This result hinted that **PL-1** and **PL-6** may induce A549 cells cycle arrest via ROS generation.

### 2.4. The Apoptotic Effects of PL Analogs against the A549 Cell

Next, we tested the apoptosis-inducing activity of PL analogs (**PL-1** and **PL-6**) on A549 cells using a flow cytometry cell apoptosis assay. As illustrated in Figure 3, **PL-1** and **PL-6** exhibited more outstanding cell death-inducing capacity against A549 cells compared to PL, and emerged in an excellent dose-dependent manner. The late apoptosis of A549 cells induced by **PL-1** (30 μM) and **PL-6** (30 μM) were 38.9% and 37.8%, respectively. Additionally, the apoptosis-inducing activities of **PL-1** (30 μM) and **PL-6** (30 μM) were remarkably reduced by pretreatment with NAC. This result also implied that **PL-1** and **PL-6** may induce cell death of A549 cells through ROS production.

### 2.5. PL Analogs Induced Reactive Oxygen Species (ROS) Generation and Redox Imbalance

Based on the Figure 3 results, we measured whether **PL-1** and **PL-6** could trigger ROS generation in A549 cells. A DCFH-DA (2′,7′-dichlorofluorescein diacetate) probe was used to test the changes of ROS levels in A549 cells. The results indicated that **PL-1** and **PL-6** remarkably increased the ROS levels compared to the control group, and displayed a dose-dependent relationship (Figure 4A). Moreover, the ROS generation abilities of **PL-1** and **PL-6** were much stronger than the leading **PL**; 30 μM **PL-6** resulted in a 2.7-fold increase of ROS levels compared to the control group. In addition, NAC could sharply decrease the ROS induced by **PL-1** (30 μM) and **PL-6** (30 μM) (Figure 4A). The ROS accumulation could cause intracellular oxidative stress. Therefore, the ratios of GSH/GSSG (Glutathione/L-Glutathione Oxidized) were measured according to the glutathione reductase-DTNB recycling assay. As shown in Figure 4B, PL analogs obviously decreased the ratios of GSH/GSSG in a dose-dependent fashion. 30 μM **PL-6** caused a 5-fold decrease of the ratio of GSH/GSSG relative to the control group. In particular, the redox imbalance induced by **PL-1** (30 μM) and **PL-6** (30 μM) were prevented significantly by NAC. These results suggest that **PL** analogs can trigger ROS accumulation and induce redox imbalance of A549 cells.

### 2.6. **PL** Analogs Induced Lipid Peroxidation and the Loss of MMP (Mitochondrial Membrane Potential)

Since the ROS accumulation could induce the lipid peroxidation of cell membrane, we measured the MDA (Malondialdehyde) levels of A549 cell after treatment with **PL** analogs. We found that both **PL-1** and **PL-6** led the increase of MDA levels (Figure 5A). The MDA levels induced by **PL-1** (30 μM) and **PL-6** (30 μM) were higher than that of **PL**. This result was in accordance with their apoptosis-inducing activities and ROS generation abilities against the A549 cells. In addition, the lipid peroxidation induced by **PL-1** (30 μM) and **PL-6** (30 μM) were fundamentally inhibited by pretreatment with NAC (Figure 5A). In addition, both **PL-1** and **PL-6** resulted in the loss of MMP in a dose-dependent manner. **PL-6** (30 μM) led to about a 10-fold decrease of MMP (mitochondrial membrane potential) compared with the control group, and was more active than that of **PL** (30 μM). The loss of MMP can be significantly blocked NAC (Figure 5B). These results indicated that **PL** analogs could trigger ROS production, then induce redox imbalance, lipid peroxidation, the loss of MMP, cell cycle arrest and A549 cell line cell death.

### 2.7. The Nucleophilic Addition Reaction of **PL** Analogs and Cysteamine

Figure 4 indicates that **PL** analogs triggered ROS generation, which can emerge through the reaction between thioredoxin reductase (TrxR) and chemotherapeutic agents containing the Michael acceptor. Therefore, we determined the electrophilicity of the Michael acceptor of **PL** analogs under pseudo-first-order conditions. As shown in Table 2, **PL-1** and **PL-6** exhibited much stronger electrophilicity than PL with *k*_2_ value of 5.70 and 8.96 M^−1^s^−1^, in line with their cytotoxicity and ROS generation ability.

In addition, we monitored the reaction of **PL-6** and cysteamine in *d_6_*-DMSO (Dimethyl-d_6_ sulfoxide) using an NMR spectrometer. As shown in Figure 6A, the doublet of doublets at δ 7.731 (H-1 and H-1’), calibrating to 2.00 hydrogen, was the protons of the aromatic nucleus. After incubating a mixture of **PL-6** and cysteamine for 1 h, the triplet integral at δ 7.332, the olefinic proton of lactam (H-3), decreased to 0.46 compared to the ^1^H-NMR of **PL-6** (Table 3). This phenomenon displayed that the Michael acceptor of lactam of **PL** analogs was the major reaction site of cysteamine. This result also hinted that the ROS generation induced by **PL** analogs was mainly due to the Michael acceptor of lactam. Moreover, after incubating **PL-6** and cysteamine for 3 days, the 1:1 adduct (*m*/*z*: 374.3040) was formed as exhibited in high-resolution mass spectrometry (HRMS) (Figure 6B).

### 2.8. The Inhibition Activity of **PL-6** on TrxR

In the the results in Figure 3 and Figure 4 and Table 3, we demonstrate that **PL** analogs reacted with cysteamine, and induced A549 cell line cells death through triggering ROS generation, which can be obtained by the inhibition of TrxR. In order to confirm the ROS production induced by **PL-6** was related to TrxR, we analyzed the expression of TrxR in A549 cells using a Western blot assay. As shown in Figure 7, the TrxR expression was decreased in a dose-dependent fashion. Moreover, the down-regulation of TrxR induced by **PL-6** can be reduced by NAC. This result was in line with the ROS generation and death of A549 cells induced by **PL-6**.

## 3. Discussion

Cancer, one of the major causes of morbidity and mortality in the world, is becoming a primary concern of human health. Chemotherapeutic agents, commonly applied in cancer diagnosis and treatment [20], could inhibit proliferation or induce cancer cell death through various molecular pathways, such as nuclear factor-κB (NF-κB) [21], vascular endothelial growth factors [22], PI3/Akt [23] and so on. In particular, the ROS, which also induce cancer cell apoptosis, can be triggered by chemotherapeutic agents containing the Michael acceptor units [16]. TrxR can be irreversibly inhibited by the Michael acceptor of chemotherapeutic agents and the modified enzyme converted into a prooxidant that triggered ROS generation [15]. **PL**, as one of the simplest alkaloids found in *Piper longum* L., contains two Michael acceptor units and has been used for the purpose of cancer prevention and treatment [17].

In this work, we obtained a panel of **PL** analogs via a nucleophilic substitution reaction between intermediate **2** and **4**, and found that the strong electrophilicity of the C2–C3 double bond was to the benefit of enhancing the cytotoxicity of **PL** analogs against cancer cells (Table 1). Moreover, the cinnamic acid unit, with strong electron-withdrawing or weak electron-donating groups, partly contribute to the anticancer activity of **PL** analogs. The results illustrated that the Michael acceptor of lactam plays a major role in the antiproliferative activity. The Michael acceptor unit can be attacked by nucleophiles, such as TrxR, GSH and so on, and results in the ROS generation [15]. TrxR of A549 cells can be inhibited by **PL-6** which also confirmed this phenomenon (Figure 7). We further identified that cysteamine mainly reacted with the Michael acceptor of lactam by the ^1^H-NMR and HRMS (Figure 6, Table 3).

The irreversible inhibition of TrxR by chemotherapeutic agents resulted in the ROS accumulation, which is linked to numerous biological processes and disease conditions [13]. Cancer cells exhibited a greater ROS level than normal cells [24]. Therefore, a common therapeutic strategy that increases ROS generation may force cancer cells to surpass the redox balance, leading to the activation of different cell death pathway [24]. For example, ROS acculumation can induce cancer cells death. As illustrated in Figure 4A, **PL** analogs displayed better activity by triggering ROS generation than that of **PL**. In addition, ROS acculumation altered the redox status, which were significantly inhibited by NAC (Figure 4B). Redox imbalance in turn influenced the production of ROS. Moreover, ROS can directly oxidize cell membrane and lead to the increase of MDA levels (Figure 5A). Generous ROS also executed cancer cell apoptosis in a mitochondrial-dependent fashion [25]. In line with this, **PL** analogs dramatically decreased the MMP (Figure 5B) and resulted in A549 cells apoptosis (Figure 3). All of the results in Figure 4 and Figure 5 were partly or entirely reversed by NAC. These results also hinted that the anticancer activities of **PL** analogs were attributed to the ROS generation.

Collectively, this work elucidates that the Michael acceptor of lactam plays a major role in **PL** analogs′ cytotoxicity. Whereas, the cinnamic acid unit, with strong electron-withdrawing or weak electron-donating groups partly contributed to that. The cysteamine mainly reacted with the C2–C3 double bond of **PL** analogs. **PL** analogs could irreversibly inhibit TrxR and result in ROS accumulation, which further disrupt redox imbalance, induce lipid peroxidation, lead to the loss of MMP, and ultimately result in A549 cell line cell death (Scheme 2).

## 4. Materials and Methods

### 4.1. Materials

Roswell Park Memorial Institute (RPMI)-1640 was from Hyclone. 3-(4,5-dimethylthiazol-2-yl)-2,5-diphenyltetrazolium bromide (MTT), rhodamine 123, 2′,7′-dichlorofluorescein diacetate, GSH and GSSG Assay kit, and thiobarbituric acid were purchased from Beyotime. The annexin V-FITC/PI apoptosis detection kit was from BD Biosciences. Substituted benzaldehyde, 2-piperidone and piperlongumine were from EnergyChemical. All other chemicals were of the highest quality available.

### 4.2. Synthesis of the PL Analogs

#### 4.2.1. The Preparation of Cinnamoyl Chloride (Intermediate **2**)

The cinnamoyl chloride **2** was prepared according to previously reported methods [17]. Briefly, substituted aldehyde (20 mmol) was condensed with malonic acid (24 mmol) in the presence of piperidine (1.2 mL) to afford a crude solid. After, it was recrystallised from ethanol to afford corresponding cinnamic acids **1**. Then, intermediate **1** (2 mmol) reacted with oxalyl chloride (6 mmol) in dry CH_2_Cl_2_ (4 mL) to produce cinnamoyl chloride **2**.

#### 4.2.2. The Preparation of 3-Chloro-5,6-dihydropyridin-2(1*H*)-one (Intermediate **4**)

The intermediate **4** was acquired from 2-piperidone according to the previously reported procedure [17]. Briefly, the mixture containing 2-piperidone (5 mmol), phosphorus pentachloride (15 mmol) and chloroform (15 mL) was refluxed for 12 h. After cooling, adjusting pH to 9, extracting with chloroform/methanol (10/1) and purifying by flash cloumn chromatography, the intermediate **3** was obtained with a white solid. The intermediate **3** (2.5 mmol) and lithium carbonate (7.5 mmol) was dissolved in DMF (Dimethyl Formamide, 5 mL) and heated to 120 °C for 12 h. After removing the solvent, the residue was extracted with chloroform and purified with cloumn chromatography to afford the intermediate **4**.

The mixture, containing intermediate **4** (2 mmol), cinnamoyl chloride **2** (2 mmol), triethylamine (6 mmol) and dry CH_2_Cl_2_ (5 mL), was stirred at room temperature for 24 h. After extracting with ethyl acetate and purification by cloumn chromatography, the target product was obtained.

(*E*)-3-chloro-1-(3-(3-methoxyl-cinnamoyl))-5,6-dihydropyridin-2(1*H*)-one (**PL-1**): Yield 16.2% (the yield is the reaction between intermediate **4** and cinnamoyl chloride **2**), yellow solid, ^1^H-NMR (500 MHz, (CDCl_3_), δ 7.74 (d, *J* = 16.0 Hz, 1H), 7.49 (d, *J* = 15.5 Hz, 1H), 7.28 (t, *J* = 8.0 Hz, 1H), 7.19 (d, *J* = 7.5 Hz, 1H), 7.09–7.11 (m, 2H), 6.93 (d, *J* = 8.0 Hz, 1H), 4.09 (t, *J* = 6.5 Hz, 2H), 3.85 (s, 3H), 2.56–2.60 (m, 2H); ^13^C-NMR (125 MHz, (CDCl_3_), δ 167.4, 160.2, 158.8, 143.5, 139.9, 135.1, 127.8, 127.2, 120.3, 120.1, 115.3, 112.1, 54.2, 40.6, 24.2. HRMS *m*/*z* (ES^+^): [M + Na]^+^ 314.0558 (theor 314.0560).

(*E*)-3-chloro-1-(3-(3-trifluoromethylphenyl)acryloyl-5,6-dihydropyridin-2(1*H*)-one (**PL-2**): Yield 12.6% (the yield is the reaction between intermediate **4** and cinnamoyl chloride **2**), yellow solid, ^1^H-NMR (500 MHz, (CDCl_3_), δ 7.81 (s, 1H), 7.74 (d, *J* = 15.5 Hz, 1H), 7.74 (t, *J* = 8.5 Hz, 1H), 7.62 (d, *J* = 8.0 Hz, 1H), 7.52 (d, *J* = 15.5 Hz, 1H), 7.49 (t, *J* = 7.5 Hz, 1H), 7.10 (t, *J* = 4.5 Hz, 1H), 4.09 (t, *J* = 6.5 Hz, 2H), 2.57–2.61 (m, 2H); ^13^C-NMR (125 MHz, (CDCl_3_), δ 168.2, 161.5, 142.5, 141.3, 135.7, 131.6, 131.4, 129.4, 128.2, 126.7, 124.9, 123.1, 122.8, 41.8, 25.3. HRMS *m*/*z* (ES^+^): [M + Na]^+^ 352.0328 (theor 352.0325).

(*E*)-3-chloro-1-(3-(3-fluorine-cinnamoyl))-5,6-dihydropyridin-2(1*H*)-one (**PL-3**): Yield 4.5% (the yield is the reaction between intermediate **4** and cinnamoyl chloride **2**), yellow solid, ^1^H-NMR (500 MHz, (CDCl_3_), δ 7.68 (d, *J* = 15.5 Hz, 1H), 7.34–7.36 (m, 2H), 7.26–7.28 (m, 2H), 7.15 (d, *J* = 15.5 Hz, 1H), 7.08–7.11 (m, 1H), 3.86 (t, *J* = 6.0 Hz, 2H), 2.84–2.86 (m, 2H); ^13^C-NMR (125 MHz, (CDCl_3_), δ 169.4, 166.3, 164.0, 143.2, 136.9, 130.4, 124.4, 121.6, 117.4, 117.2, 114.8, 114.6, 45.4, 20.3. HRMS *m*/*z* (ES^+^): [M + Na]^+^ 302.0360 (theor 302.0358).

(*E*)-3-chloro-1-(3-(4-nitro-cinnamoyl))-5,6-dihydropyridin-2(1*H*)-one (**PL-4**): Yield 7.1% (The yield is the reaction between intermediate **4** and cinnamoyl chloride **2**), yellow solid, ^1^H-NMR (500 MHz, (CDCl_3_), δ 8.23 (d, *J* = 8.5 Hz, 2H), 7.72–7.75 (m, 3H), 7.56 (d, *J* = 16.0 Hz, 1H), 7.12 (t, *J* = 4.5 Hz, 1H), 4.10 (t, *J* = 6.0 Hz, 2H), 2.59–2.62 (m, 2H); ^13^C-NMR (125 MHz, (CDCl_3_), δ 167.8, 161.5, 148.5, 141.5, 141.1, 140.9, 128.9, 128.1, 125.5, 124.1, 41.8, 25.3. HRMS *m*/*z* (ES^+^): [M + Na]^+^ 329.0304 (theor 329.0305).

(*E*)-3-chloro-1-(3-(4-trifluoromethyl-cinnamoyl))-5,6-dihydropyridin-2(1*H*)-one (**PL-5**): Yield 3.7% (the yield is the reaction between intermediate **4** and cinnamoyl chloride **2**), yellow solid, ^1^H-NMR (500 MHz, (CDCl_3_), δ 7.73 (d, *J* = 15.5 Hz, 1H), 7.66 (t, *J* = 8.5 Hz, 2H), 7.62 (t, *J* = 8.5 Hz, 2H), 7.53 (d, *J* = 15.5 Hz, 1H), 7.11 (t, *J* = 4.5 Hz, 1H), 4.10 (t, *J* = 6.5 Hz, 2H), 2.58–2.61 (m, 2H); ^13^C-NMR (125 MHz, (CDCl_3_), δ 167.0, 160.3, 141.2, 140.2, 137.1, 127.3, 127.0, 124.6, 122.6, 121.7, 40.6, 28.6, 24.2. HRMS *m*/*z* (ES^+^): [M + Na]^+^ 352.0325 (theor 3352.0328).

(*E*)-3-chloro-1-(3-(4-fluorine-cinnamoyl))-5,6-dihydropyridin-2(1*H*)-one (**PL-6**): Yield 7.1% (the yield is the reaction between intermediate **4** and cinnamoyl chloride **2**), yellow solid, ^1^H-NMR (500 MHz, (CDCl_3_), δ 7.71 (d, *J* = 15.5 Hz, 1H), 7.57-7.59 (m, 2H), 7.42 (d, *J* = 15.5 Hz, 1H), 7.09 (t, *J* = 4.5 Hz, 1H), 7.05 (t, *J* = 8.5.5 Hz, 1H), 4.08 (t, *J* = 6.5 Hz, 2H), 2.56-2.59 (m, 2H); ^13^C-NMR (125 MHz, (CDCl_3_), δ 168.4, 165.0, 161.5, 143.3, 141.1, 131.2, 130.3, 128.3, 121.0, 116.0, 41.7, 25.3. HRMS *m*/*z* (ES^+^): [M + Na]^+^ 302.0358 (theor 302.0360).

### 4.3. Cytotoxicity Assay

The cytotoxicity of **PL** analogs against cancer cells was evaluated by the MTT assay [26]. In brief, A549 (3 × 10^3^/well) and SK-OV3 (5 × 10^3^/well) cells were treated with PL analogs for 48 h before measuring.

### 4.4. Stability Test

The stability of **PL** analogs in PBS buffer (100 mM) at 25 °C was surveied as previously reported [27]. Briefly, we observed the changes of maximun absorbance of the test compounds for 120 min at 10-min intervals.

### 4.5. Cell Cycle Analysis

After incubation with the test compounds for 15 h, A549 cells were collected, dyed with PI, and analyzed by flow cytometry as previously reported [26].

### 4.6. Cancer Cells Apoptosis Assay

After treatment with the test compounds for 18 h, A549 cells were gathered, stained with FITC Annexin-V and PI, and analyzed with a flow cytometry as previously reported [28].

### 4.7. ROS Detection

A549 (3 × 10^5^) cells were incubated with the tested compounds for 6 h, then, the cells were collected, stained with DCFH-DA for 30 min, and subsequently analyzed by flow cytometry as described previously [28].

### 4.8. The Evaluation of Intracellular Redox Balance Levels

After treatment with the test compounds for 6 h, the cells were harvested, lysed, and assayed for determining the ratio of GSH/GSSG as previously reported [26].

### 4.9. Determination of Lipid Peroxidation

A549 cells (3 × 10^5^ cells/well) were treated with the test compounds for 15 h before collecting. After lysing in RIPA buffer (RIPA Lysis Buffer), the supernatant was used to determine the lipid peroxidation as previously described [27].

### 4.10. Measurement of Mitochondrial Membrane Potential

Incubation with the test compounds for 15 h, the cells were harvested, stained with Rhodamine 123, and analyzed by flow cytometry as previously reported [27].

### 4.11. Electrophilicity Assessment by a Kinetic Thiol Assay

The electrophilicity of **PL** analogs were evaluated by the pseudo-first-order assay and the results were expressed second-order rate constants (*k*_2_) as previously reported [29]. Briefly, the test compounds (50 μM) were mixed with 100-fold cysteamine at 25 °C, and monitored the decay of their wavelength maxima by using an ultraviolet–visible (UV-Vis) spectrophotometer. The second-order rate constants (*k*_2_) were obtained by plotting the pseudo-first-order constants vs. [cysteamine].

### 4.12. Determination of the Reaction Site by Nuclear Magnetic Resonance (^1^H-NMR) and High-Resolution Mass Spectrometry (HRMS)

^1^H-NMR spectra of **PL-6** (30 mM in *d*_6_-DMSO) after adding cysteamine (90 mM in *d_6_*-DMSO) at set intervals (0, 1 h) were recorded using a Bruker AV 500 spectrometer. After incubation 3 days, the reaction products were monitored by HRMS (Acquity UPLC I-Class/Xevo G2-XS QTOF) [17].

### 4.13. Western Blot Analysis

Briefly, after treatment with the test compounds for 18 h, A549 cells were collected and lysed with ice-cold RIPA lysis buffer containing proteinase inhibitors. The expression of TrxR proteins after treatment with the test compounds was analyzed by Western blot as previously described [30].

### 4.14. Statistical Analysis

The data are expressed as the mean ±S.D. of at least three independent experiments.

Data were analyzed using one-way analysis of variance (ANOVA) followed by the Dunnett test (comparing all drug treatment groups vs. control). The values were considered significant at *p* < 0.05. *: *p* < 0.05; **: *p* < 0.01; ***: *p* < 0.001 compared with control.

## Data Availability

The data of the ^1^H-NMR, ^13^C-NMR and HRMS of **PL** analogs are available in the Appendix A.

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
