# Peer review of "Piperlongumine Analogs Promote A549 Cell Apoptosis through Enhancing ROS Generation"

_molecules, 2021, doi:10.3390/molecules26113243_

Round 1

Reviewer 1 Report

The manuscript submitted to me for review and evaluation describes the synthesis of 6 new compounds that are derivatives of Piperlongumine (PL) and presents a wide set of biological research results conducted mainly on A549 and SK-OV3 cell lines. These studies were aimed at showing the cytotoxicity of the obtained compounds and at an attempt to explain mechanism of its activity. The obtained results illustrated that the Michael acceptor of lactam structure play a major role on the antiproliferative activity. Also, chlorine atom at C-2 position playing main role in the enhance cytotoxicity of PL analogues.

The authors come to interesting conclusions, which are well documented with numerous experimental materials.

In addition to the general particularly good opinion about presented article, unfortunately I have also a few minor formal, editorial and technical comments.

The first ones include:

- The article, not only in terms of content, but also in terms of form, is terribly like other papers of the same team, quoted here as [26, 28], and is also like article referenced as [17].

- Scheme 2 in the assessed article is terribly like the scheme in [28].

- All biological studies are described with the words "as previously reported", and this reference is made to authors own previous works.

These three remarks do not require correction, but they result in a lower evaluation of the work in terms of its innovativeness and novelty.

While other more technical comments include:

- There are many places where the word "band" is used instead of the word "bond" which should appear there as correct word.

- Page 3 line 78, the symbol of Celsius degrees should be written more correctly.

- Page 5 line 152, title of Table 2; is it really about curcumin? after all, this work does not concern it at all.

- Page 6 Fig. 6. For me, these spectra are completely unreadable. They should be either enlarged or edited in black as it is in the structure of the molecule in the NMR spectrum.

- The authors repeatedly refer to the fact that the described compounds are derivatives of cinnamic acid, they write that they were obtained from its substituted derivatives. However, in key places, especially when describing the obtained products (but not only), they use the name derived from acrylic acid. In my opinion, such nomenclature, although also correct, may seem a bit confusing to the reader. I admit that the name used by the authors is easier to create and it has already been used in the published work [17]. For me, however, it would be more convincing to derive the names from cinnamic acid, especially since in many places this tendency appears in a more general form in all this article.

- It should be specified for which substrate the reaction yield is calculated, because it is quite low. Does it concern all reaction way or only the last step (2 + 4)?

After introducing these corrections and additions, I suggest accepting the reviewed work for edition in your Journal without repeated reviewing process.

Author Response

Dr. Kayla Li

Assistant Editor

Molecules

May 19, 2021

Dear Dr. Kayla Li,

Thank you very much for overseeing the review of our revised manuscript (molecules-1201252). Those comments are highly insightful and enabled us to greatly improve the quality of our manuscript. We have studied comments carefully and tried our best to address each of the reviewer's comments through changes to our revised manuscript and Supporting Information. Revised portion is marked in red in the paper. The main corrections in the paper and the responds to the reviewer’s comments are as following:

  • Reviewer #1, comment #1: There are many places where the word "band" is used instead of the word "bond" which should appear there as correct word.

Thanks for your careful reading of our manuscript. We had carefully checked and

corrected the word in our revised manuscript. The revised portions were marked in red in our revised manuscript.

  • Reviewer #1, comment #2: Page 3 line 78, the symbol of Celsius degrees should be written more correctly.

Thanks for your careful reading of our manuscript. We had corrected the symbol of Celsius degrees. The revised portions were marked in red in our revised manuscript.

  • Reviewer #1, comment #3: Page 5 line 152, title of Table 2; is it really about curcumin? after all, this work does not concern it at all..

Thanks for your careful reading of our manuscript. We had corrected “curcumin” to “PL” in our revised manuscript. The revised portions were marked in red in our revised manuscript.

  • Reviewer #1, comment #4: Page 6 Fig. 6. For me, these spectra are completely unreadable. They should be either enlarged or edited in black as it is in the structure of the molecule in the NMR spectrum.

Thanks for your valuable advice. We has enlarged and edited the Fig. 6 in black.

  • Reviewer #1, comment #5: The authors repeatedly refer to the fact that the described compounds are derivatives of cinnamic acid, they write that they were obtained from its substituted derivatives. However, in key places, especially when describing the obtained products (but not only), they use the name derived from acrylic acid. In my opinion, such nomenclature, although also correct, may seem a bit confusing to the reader. I admit that the name used by the authors is easier to create and it has already been used in the published work [17]. For me, however, it would be more convincing to derive the names from cinnamic acid, especially since in many places this tendency appears in a more general form in all this article.

Thanks for your valuable advice. We have corrected the compounds as cinnamic acid derivatives in our revised manuscript. The revised portion is marked in red in the paper.

  • Reviewer #1, comment #6: It should be specified for which substrate the reaction yield is calculated, because it is quite low. Does it concern all reaction way or only the last step (2 + 4)?.

Thank you very much. The reaction yield, which has been specified for in our revised manuscript, concerns only the last step (between intermediate 4 and cinnamoyl chloride 2). The revised portion is marked in red in the paper.

The resulting revised manuscript and Supporting Information have been submitted online. We are optimistic that the attached revised manuscript will be a valuable and interesting contribution to Molecules.

Thank you very much for your time and assistance. Please contact me ([email protected]) if you require any additional information. I look forward to hearing from you soon.

Sincerely yours,

                                  Dr. Guo-Yun Liu

                 School of Pharmaceutical Sciences, Liaocheng University

Reviewer 2 Report

The Turnitin Similarity Index is high, the authors need to reduce it.

The abstract should be proof-read. The cancer apoptosis should be referred to as specific cancer cells.

The manuscript should English-edited, especially the introduction. Too many typos and grammatical errors or sentence reconstructions. 

There is a difference between medicine and a medicinal plant.

Why do the authors want to make modifications to PL while it has been shown to possess anticancer activities against several other cancer cells, especially, through generation of ROS?

The images resolutions should be improved.

The FACS data presented in Figure 2 and 3 should also be plotted to incorporate the repeats.

The authors should refer to figures not "above results".

Author Response

Dr. Kayla Li

Assistant Editor

Molecules

May 19, 2021

Dear Dr. Kayla Li,

Thank you very much for overseeing the review of our revised manuscript (molecules-1201252). Those comments are highly insightful and enabled us to greatly improve the quality of our manuscript. We have studied comments carefully and tried our best to address each of the reviewer's comments through changes to our revised manuscript and Supporting Information. Revised portion is marked in red in the paper. The main corrections in the paper and the responds to the reviewer’s comments are as following:

  • Reviewer #2, comment #1: The Turnitin Similarity Index is high, the authors need to reduce it.

Thanks for your valuable comments. We have tried our best to correct the manuscript, and the revised portion is marked in red in the paper.

  • Reviewer #2, comment #2: The abstract should be proof-read. The cancer apoptosis should be referred to as specific cancer cells.

Thanks for your valuable comments. The abstract has been proof-read and the cancer apoptosis has corrected as specific cancer cells death in our revised manuscript. The revised portion is marked in red in the paper.

  • Reviewer #2, comment #3: The manuscript should English-edited, especially the introduction. Too many typos and grammatical errors or sentence reconstructions.

Thanks for your advice. The English of the manuscript has been edited in our revised manuscript. The revised portion is marked in red in the paper.

  • Reviewer #2, comment #3: There is a difference between medicine and a medicinal plant.

Thanks for your advice. We has deleted the “medicinal plant” expression in our revised manuscript.

  • Reviewer #2, comment #4: Why do the authors want to make modifications to PL while it has been shown to possess anticancer activities against several other cancer cells, especially, through generation of ROS?

Thanks for your valuable comments. Piperlongumine (PL) is well known to possess multiple pharmacological activity including as anti-microbial, anti-parkinsonian, anti-epileptic and anticancer activity. Of particular interest is the anticancer property of PL, which could induces cancer cells apoptosis through generation of ROS. Yan and co-workers have design piperlongumine-directed anticancer agents by an electrophilicity-based prooxidant strategy [Free Radic.Biol. Med., 2016, 97: 109-123]. The structure-activity relationship of PL and its analogs reveals that C2-C3 double bond plays a critical role on the ROS generation and glutathione lessening. Whereas, C7-C8 double bonds is partial correlation with the antiproliferative activity of PL, and the effect of the pharmacophore on anticancer activity still need to be explored. In addition, the Michael acceptor of chemotherapeutic agents can irreversibly inhibit TrxR and trigger intracellular ROS accumulation. Therefore, in order to find the excellent anticancer drug and explore the effects of electronic effect of substituents on the cytotoxicity, we design and synthesized a panel of PL analogs with electron-withdrawing group (-F and CF3) or electron-donating group (-OCH3) attaching to the aromatic ring. Next, we assessed the electrophilicity of PL analogs and explored the anticancer mechanism.

  • Reviewer #2, comment #5: The images resolutions should be improved.

Thanks for your valuable comments. The images resolutions has been improved in our revised manuscript.

  • Reviewer #2, comment #6: The FACS data presented in Figure 2 and 3 should also be plotted to incorporate the repeats.

Thanks for your valuable comments. The FACS data presented in Figure 2 and 3 have been plotted to incorporate the repeats in the revised manuscript.

Figure 2. The effect of PL analogs (PL-1 and PL-6) on the cell cycle arrest of A549 cells at the indicated concentrations in the presence or absence of NAC. *P < 0.05, **P < 0.01, ***P < 0.001 compared with control.

Figure 3. The effects of PL analogs (PL-1 and PL-6) on the specific cancer cells death-inducing activity of A549 cells at the indicated concentrations in the presence or absence of NAC. *P < 0.05, **P < 0.01, ***P < 0.001 compared with control.

  • Reviewer #2, comment #7: The authors should refer to figures not "above results".

Thanks for your valuable comments. We have corrected it in our revised manuscript.

The resulting revised manuscript and Supporting Information have been submitted online. We are optimistic that the attached revised manuscript will be a valuable and interesting contribution to Molecules.

Thank you very much for your time and assistance. Please contact me ([email protected]) if you require any additional information. I look forward to hearing from you soon.

Sincerely yours,

                                  Dr. Guo-Yun Liu

                 School of Pharmaceutical Sciences, Liaocheng University

Round 2

Reviewer 2 Report

Line 16: Change "play" to "plays". 

Line 19: The sentence should read as follows:

Then, the ROS accumulation could disrupt the redox balance, induce lipid peroxidation, lead the loss of MMP, and ultimately result in cell cycle arrest and A549 cell line cell death.

Line 21: The authors demonstrated their findings in vitro, therefore, they should state so.

In conclusion, PL analogs could induce in vitro cancer apoptosis through the inhibition of TrxR and ROS accumulation.

Line 33: The sentence should read as follows:

Moreover, PL induces cell cycle arrest and death of A549 cells through triggering ROS accumulation in cancer cells[11,12].

Line 35: The sentence should read as follows:

ROS, well known as an upstream signal involved in the induction of cancer cell death, can be triggered by several exogenous and endogenous factors[13].

The following sentence does not make sense, it needs to be rephrased:

Chemotherapeutic agents, a classic exogenous factor, enhance ROS reaching a toxic threshold to encourage specific cancer cells death [14].

Line 47-8 should be revised as follows:

Whereas, C7-C8 double bonds is partial correlation with the anti-proliferative activity of PL, and the effect of the pharmacophore on anticancer activity remains unexplored.

Lines 100 to 104: The sentences should read as follows:

As illustrated in Fig. 3, PL-1 and PL-6 exhibited more outstanding cell death inducing capacity against A549 cells compared to PL, and emerged as an excellent dose-dependent manner. The late apoptosis of A549 cells induced by PL-1 (30 μM) and PL-6 (30 μM) were 38.9% and 37.8%, respectively. Additionally, the apoptosis inducing activities of PL-1 (30 μM) and PL-6 (30 μM) on A549 cells were remarkably reduced by pretreatment with NAC.

Line 105 should read as follows:

This result also implied that PL-1 105 and PL-6 may induce cell death of A549 cells through ROS production.

The first line of figure 3 legend should read as follows:

Figure 3. The apoptotic effects of PL analogs (PL-1 and PL-6) against the A549 cells at the indicated concentrations in the presence or absence of 109 NAC.

On my previous comment, I meant that the authors should be specific when they state induction of apoptosis. They need to state the cells that the are referring to not change apoptosis to specific cells death. I see that they have written this throughout the manuscript. That should be corrected.

Author Response

Molecules

May 25, 2021

Dear Reviewer,

Thank you very much for overseeing the review of our revised manuscript (molecules-1201252). Those comments are highly insightful and enabled us to greatly improve the quality of our manuscript. We have tried our best to address each of the reviewer's comments and marked up using the “Track Changes” function in our revised manuscript. The main corrections in the paper and the responds to the reviewer’s comments are as following:

  • Reviewer #2, comment #1: Line 16: Change "play" to "plays".

Line 16: We have change "play" to "plays" in our revised manuscript.

  • Reviewer #2, comment #2: Line 19: The sentence should read as follows:

Then, the ROS accumulation could disrupt the redox balance, induce lipid peroxidation, lead the loss of MMP, and ultimately result in cell cycle arrest and A549 cell line cell death.

Line 19: We have corrected the sentence as the reviewer’s comment.

  • Reviewer #2, comment #3: Line 21: The authors demonstrated their findings in vitro, therefore, they should state so.

Line 21: We have state “in vitro” in line 21 of our revised manuscript.

  • Reviewer #2, comment #4: Line 33: The sentence should read as follows: Moreover, PL induces cell cycle arrest and death of A549 cells through triggering ROS accumulation in cancer cells[11,12].

Line 33: We have corrected the sentence as the reviewer’s comment.

  • Reviewer #2, comment #5: Line 35: The sentence should read as follows:

ROS, well known as an upstream signal involved in the induction of cancer cell death, can be triggered by several exogenous and endogenous factors[13].

Line 35: We have corrected the sentence as the reviewer’s comment.

  • Reviewer #2, comment #6: The following sentence does not make sense, it needs to be rephrased:

Chemotherapeutic agents, a classic exogenous factor, enhance ROS reaching a toxic threshold to encourage specific cancer cells death [14].

Thank you very much. We have deleted the sentence in our revised manuscript.

  • Reviewer #2, comment #7: Line 47-8 should be revised as follows:

Whereas, C7-C8 double bonds is partial correlation with the anti-proliferative activity of PL, and the effect of the pharmacophore on anticancer activity remains unexplored.

Line 47-8: We have revised the sentence as the reviewer’s comment in our revised manuscript.

  • Reviewer #2, comment #8: Lines 100 to 104: The sentences should read as follows:

As illustrated in Fig. 3, PL-1 and PL-6 exhibited more outstanding cell death inducing capacity against A549 cells compared to PL, and emerged as an excellent dose-dependent manner. The late apoptosis of A549 cells induced by PL-1 (30 μM) and PL-6 (30 μM) were 38.9% and 37.8%, respectively. Additionally, the apoptosis inducing activities of PL-1 (30 μM) and PL-6 (30 μM) on A549 cells were remarkably reduced by pretreatment with NAC.

Lines 100 to 104: We have corrected these sentences as the reviewer’s comment in our revised manuscript.

  • Reviewer #2, comment #9: Line 105 should read as follows:

This result also implied that PL-1 and PL-6 may induce cell death of A549 cells through ROS production.

Lines 105: We have corrected these sentences as the reviewer’s comment in our revised manuscript.

  • Reviewer #2, comment #10: The first line of figure 3 legend should read as follows:

Figure 3. The apoptotic effects of PL analogs (PL-1 and PL-6) against the A549 cells at the indicated concentrations in the presence or absence of NAC.

We have corrected the figure 3 legend as the reviewer’s comment in our revised manuscript.

  • Reviewer #2, comment #11: On my previous comment, I meant that the authors should be specific when they state induction of apoptosis. They need to state the cells that the are referring to not change apoptosis to specific cells death. I see that they have written this throughout the manuscript. That should be corrected.

    Thank you very much. We have corrected these mistakes in our revised manuscript.

The resulting revised manuscript have been submitted online. We are optimistic that the attached revised manuscript will be a valuable and interesting contribution to Molecules.

Thank you very much for your time and assistance. Please contact me ([email protected]) if you require any additional information. I look forward to hearing from you soon.

Sincerely yours,

                                  Dr. Guo-Yun Liu

                 School of Pharmaceutical Sciences, Liaocheng University
